# The programming curriculum within ISIS

**Marion Deslandes-Martineau**[1☯]*, **Patrick Charland**[1☯]*, **Hugo G. Lapierre**[1],
**Olivier Arvisais**[1], **Chirine Chamsine**[1], **Vivek Venkatesh**[2], **Mathieu Guidère**[3]

**1** UNESCO Chair in Curriculum Development (UCCD), Department of Didactics, Université du Québec à Montréal (UQAM), Montreal, Canada, **2** UNESCO co-Chair in Prevention of Radicalisation and Violent Extremism, Department of Art Education, Concordia University, Montreal, Canada, **3** Department of Languages, National Institute of Health and Medical Research (INSERM), Université Paris 8, Paris, France

☯ These authors contributed equally to this work.
* deslandes-martineau.marion@courrier.uqam.ca (MDM); charland.patrick@uqam.ca (PC)

**Data Availability Statement:** PLOS' data availability policy (https://journals.plos.org/plosone/s/data-availability) specifies that in some

## Abstract

From 2014 to 2017, the Islamic State in Irak and Syria (ISIS), a terrorist political organization of Salafist jihadist ideology, had put in place an operational and relatively stable educational system. Among its Complementary Programs, ISIS included a curriculum for programming using the Scratch software. In this article, we discuss this curriculum by analyzing the content of the official ISIS programming textbook, with the objectives of characterizing: 1) the curriculum's pedagogical intentions and definition of programming; 2) the programming curriculum; and 3) the religious and military indoctrination value. We found that, first, ISIS's programming curriculum intentions are more about religious and military injunctions to build the caliphate than they are about developing 21st-century skills such as computational thinking. Second, although the progression of learning in the sequence of activities designed by ISIS seems logical and, overall, well-ordered, the ISIS programming curriculum does not encourage the development of 21st-century skills such as problem solving, discovery learning, or creativity—nor for that matter, the transfer of programming knowledge to different contexts. Finally, the textbook is particularly rich in elements of military and religious indoctrination and effectively participates in the indoctrination of students by helping to inculcate values consistent with ISIS's jihadist ideology. This contribution seeks to better understand ISIS's approach to education, which could provide support for initiatives aimed at rebuilding impacted education systems and groups.

## 1. Introduction

Many are unaware that the Islamic State in Iraq and Syria (ISIS), a terrorist political organization of Salafist jihadist ideology, has established an institutionalized and functional educational system in Iraq and Syria [1]. Indeed, ISIS has developed a fairly elaborate curriculum and a substantial amount of educational material in a short period of time [1]. From 2014 to 2017, it is estimated that nearly 150,000 students attended ISIS schools, or about one-third of school-age youth living under the regime [2, 3].

instances, authors may not be able to make their underlying data set publicly available for legal or ethical reasons. This paper relies on Islamic State physical school books that were sent from local NGOs (third-party data) to the research team. For staff security purposes, our partner (the NGO) asked that the documents not be shared. Given the ethical stakes that arise when it comes to fully sharing school manuals that put forward violent and hateful discourses and the potential security implications for sharing contact information for the NGO involved in the project, the data for this article are not available.

**Funding:** CC: This study received a financial support from the Social Science and Humanities Research Council (SSHRC) – Insight Development Grant (file number 430-2019-00879). The funders had no role in study design, data collection and analysis, decision to publish, or preparation of the manuscript. https://www.sshrc-crsh.gc.ca/.

**Competing interests:** The authors have declared that no competing interests exist.

The educational materials designed and disseminated by ISIS, including textbooks, provide a relevant gateway to better understand their approach to education and the various disciplines within this totalitarian regime. The ISIS curriculum is divided into two blocks: Fundamental Programs (Doctrine, Quran, Tradition, Life of the Prophet, Calligraphy, Arabic, Islamic Education, and Physical Education) and Complementary Programs (science, mathematics, physics, chemistry, history, geography, computer science, and English). Interestingly, despite the pruning of several mainstream school subjects such as art or music in the ISIS curriculum, computer science—specifically introduction to programming—was introduced as a compulsory subject for the first year of secondary school (Grade 6) [4]. The integration of computer science and technology is a staple of modern curricula [5, 6], and ISIS's is no exception. Indeed, in the introduction to its programming textbook, ISIS emphasizes that programming skills are central skills for using and developing modern technologies in times of peace and war and for ensuring technical and scientific autonomy for the society.

In this article, we will analyze an introductory programming textbook based on the Scratch software and intended for 6th-grade students. To properly frame this analysis, we will first present the history of the Islamic State in Iraq and Syria, its ideology, and the establishment of its educational system. We will then present our theoretical framework, including the concepts of curriculum, textbook, and religious and military indoctrination, as well as the research approach we have undertaken to analyze the textbook.

Note that some of the work cited in this article is the work of the authors, i.e., references 2, 4, 8, and 13. Furthermore, references 2, 4, and 13 are, like this article, part of the UNESCO Chair in Curriculum Development (UCCD) project on education in the Islamic state.

## 1.1 The rise of the Islamic State

The Islamic State is a radical terrorist organization that emerged during the Iraqi Civil War between 2003 and 2011, but whose beginnings are believed to date back to the 1980s [7]. Between 2004 and 2006, the armed group first known as Al-Qaeda in Iraq was at the heart of the jihadist resistance against the American occupation [8]. After the death of Al-Qaeda's leader in 2006, jihadist leaders united to form the Islamic State. Over the years, ISIS's armed jihadist groups (led by Abu Bakr al-Baghdadi from 2010 until his death in 2019) conquered several territories in Iraq and Syria, capitalizing in 2011 on the withdrawal of American troops from Iraq and the civil war in Syria to expand their influence [4]. In June 2014, after the capture of the city of Mosul, Iraq's second-largest city, al-Baghdadi declared the abolition of official borders between Iraq and Syria and the establishment of a caliphate. By 2015, at the height of its power, ISIS was a proto-state the size of Great Britain, with a highly developed bureaucracy and several ministries in place, including a Ministry of Education [9]. After 2015, the influence of ISIS declined, until its dismantlement in December 2017. To date, only a small area of Syria remains under ISIS's control.

According to Olidort [1], ISIS is characterized by a Salafist jihadist ideology. Salafism is a Sunni fundamentalist movement adhering to a very strict theological and legal view of Islam based on the beliefs and practices of the early Salafists—the first companions of the Prophet Muhammad. In defense of this "good version" of Islam, ISIS advocates jihad (holy war), one facet of which is the "Greater Jihad," which Sunni Muslims must wage against external non-Muslim enemies—notably the disbelieving West—and another facet of which is the "Lesser Jihad," which they must wage against internal enemies—notably non-Sunni Muslims, the heretics [8] (authors' work). It is important to note here that while "jihad" can have many meanings, we will only retain the activist meaning exclusively considered by ISIS, that of holy war

against all enemies of Muslims, and the application of a primitive Salafi version of Sharia law [2, 8] (authors' work).

ISIS is also characterized by extreme violence, especially toward children [4], and has all the characteristics of a totalitarian regime according to Freidrich and Brzezinski [10]: a regime with a single party led by a charismatic leader (al-Baghdadi), a strong and exclusive ideology, a police force based on a regime of terror, and total control of the economy and mass communication. In addition to exploiting children for war and abusing them, thus making them both participants and victims of the regime [11], ISIS has implemented a significant campaign of child indoctrination via a relatively well-institutionalized educational system. Indeed, it is easy to recognize the phases of children's involvement in terrorism (structural and predatory recruitment; enlistment; pre-training indoctrination; training; and deployment) [12] in the actions taken by ISIS. Horgan et al. [11] even point out that ISIS's indoctrination effort is even more elaborate, structured, and effective. As a totalitarian regime, ISIS had all the authority between 2014 and 2017 to discard existing educational systems and implant one that conformed to its extremist religious positions in all areas under its control [2, 13]. Thus, education within this reformed system is heavily influenced by Salafi jihadism—a religious doctrine marked by militarism and extreme violence.

## 1.2 Establishing an educational system within ISIS

In August 2014, the appointed Minister of Education Dhu al-Qarnayn's new "Curriculum Office" issued a series of decisions prior to official educational reform, including the removal of certain subjects, the prohibition of any reference to patriotism or nationalism (in favor of exclusive belonging to Islam and the Muslim community), the removal of examples referring to democracy or elections from mathematics courses, and the removal of Darwin's evolutionary theory in favor of creationism along with the attribution of all natural, chemical, or physical laws to Allah [4]. In November 2014, ISIS reopened schools and "started over" by erasing all traces of educational systems "corrupted by Western influence" to rely exclusively on the doctrine of early Salafists [4]. The official *Education Policy in the Islamic Caliphate State* states that in "an era of increased falsification, fakery, hatred, and outrageousness," the Islamic education provided by ISIS is based on "a clear vision, neither Eastern nor Western, but prophetic Quranic that moves away from the passions, heresies, and impostures emanating from the missionaries of Eastern socialism and Western capitalism or the agents of vicious parties and movements in every corner of the world."

The production of new textbooks was set in motion at the same time, and these were prepared in record time (less than a year) by committees following the main instruction to simplify and "Islamize" all content. Textbooks for several religious disciplines, but also for writing, reading, chemistry, biology, English, physical education, and programming were quickly made available in digital and paper versions [1, 4].

It is worth noting that at the time of the ISIS takeover, the Iraqi and Syrian educational systems had already been damaged by several years of war, attacks, and bombings, as well as by the exile of many teachers [4]. The power takeover by ISIS in June 2014 thus marked the beginning of a very rapid and methodical shift to a unified educational system. The educational reform implemented rapidly and massively by ISIS is a unique case in contemporary history, and a large body of educational materials published by the organization provides a comprehensive view of this major indoctrination campaign [11]. Not only are these textbooks the only educational materials allowed and used for teaching under the ISIS educational reform, but they are also filled with doctrinal elements that are hard to ignore [4].

### 1.3 Religious and military references

For Bourdieu and Passeron [14], the educational system systematizes social reproduction, meaning that it transmits social and cultural values and norms from generation to generation. One of the roles of education is thus to reproduce culture and offer a representation of the ideal citizen in a given society [2]. Education has the potential to shape certain ideas and behaviors in citizens and to orient them toward predetermined principles and values. It is not surprising, then, that ISIS's educational system draws references from a highly militarized jihadist doctrine.

More than half of the school schedule is allocated to religious studies (the memorization of the Qur'an being at the basis of their educational system) and even "secular" subjects, such as chemistry, biology, English, physical education, or programming are in the service of religious indoctrination or preparation for jihad [13]. Religious extremism is of course present in religion textbooks, but textbooks from all other disciplines are also riddled with religious passages, justifying, to varying degrees, the use of violence to achieve ISIS's goals [1]. As for the specific curriculum in programming, it is intimately linked to religious activities since it is on Fridays—an important day of prayer—that students meet for team projects. Religious and computer science assignments are thus closely intertwined [13].

Religion instrumentalizes the different school subjects within this system, which in turn feed into ISIS's political agenda [2]. Olidort [1] identifies two levels of ISIS's curriculum organization. On the one hand, a "Salafization" of Islam in religion textbooks defines the Islamic beliefs and practices advocated by ISIS; on the other hand, an "ISization" of society in textbooks of other disciplines contributes to the development of "ideal citizens" for ISIS. Since religious doctrine is central and intrinsic to the curriculum, ISIS's action-oriented religious ideology strongly influences the educational objectives, which are then more political than pedagogical [13].

In terms of military references, the educational reform put in place by ISIS specifically states that jihadist teachings must be intensified in school and include, among other things, the handling of weapons and warlike strategies. Many such military references fill the ISIS curriculum. For example, the physical education curriculum includes elements of military training such as assembling firearms, and the textbook cover features students standing in an arena wearing combat gear [1, 4]. Military ideology is also prominent in several other ISIS textbooks, including the mathematics textbook, where military elements are often used as a starting point to contextualize abstract learning (e.g., the students learn to count ammunition) [1, 13]. Many textbooks are also replete with numerous iconographic elements related to militarism (images of weapons, planes, tanks, missiles, etc.).

### 1.4 References in curricular modernity

ISIS's educational system also draws references from curricular modernity, with several aspects reflecting rather contemporary approaches in education. For instance, according to Arvisais and Guidère [13] (authors' work), the curriculum implemented by ISIS is in line with an integrated approach to education. This type of approach has been one of the main orientations in education since at least the turn of the century and can take many forms [15], but it generally leaves religion out of the curriculum structure [13]. However, in religiously controlled states, religion plays a central role in the development, implementation, and evaluation of educational programs [16]. This trend is taken to the extreme when radical organizations are in power [13]. Under the rule of ISIS, the core of the curriculum resides in a version of Salafi jihadism that conforms to the organization's interests and promotes the establishment of a caliphate through the use of extreme violence [1].

In addition to the integrative approach in its curriculum, ISIS's educational system places overall importance on information and communication technologies (ICT). Indeed, all educational materials developed by ISIS have been widely disseminated in digital versions on the Internet and social networks [4], and mobile apps have been developed by ISIS for learning the alphabet and supplications [1]. Furthermore, despite the pruning of several school subjects such as art, music, or philosophy [4], ISIS has chosen to include computer science in its curriculum, specifically via programming learning.

**1.4.1 Learning programming within ISIS.**   According to Romero [6], today's societies need learners capable of innovation, creativity, and adaptation to technological environments. Modern education can no longer do without teaching digital literacy, media, ICT, and other technology-related skills [5]. Programming is a "next-generation discipline" in that it involves a 21st-century skill: solving increasingly complex problems [5, 6].

Recently, and in line with these goals, some educational systems, such as the ones in Estonia, Australia, the United Kingdom, and, closer to home, British Columbia, have made learning programming mandatory for all their students. Programming classes may begin very early. In some cases, students as young as five years old are asked to program computers [17]. Similarly, in the introduction to their programming textbook [18], ISIS states that its mission is to prepare and qualify students for the use of modern technologies and to develop their programming skills, independence, and creativity in science and technology.

However, learning to program is a long and difficult process, even for adult university learners [19, 20]. This learning process can be even more complex for young learners for three reasons: 1) traditional programming languages are not grade-appropriate for young students, since their syntax is far removed from natural language [21]; 2) learning programming is challenging for young learners, since it requires some capacity for abstraction in order to see the purpose of a program; and 3) the feedback offered by traditional programming software is limited and not well grounded in the physical reality of young learners [22].

To limit these impediments to learning programming, many techno-pedagogical solutions have been developed over the past 15 years. These solutions are generally grouped into two main categories. Programming learning among young people has been facilitated, on the one hand, by the development of tools using visual programming languages which reduce the necessary prerequisite knowledge in mathematics [23] and, on the other hand, by the use of feedback tools, such as simulation software [24], which offer more concrete feedback and allow for a better contextualization of the abstract learning done in programming.

ISIS has thus built a programming curriculum for youth based on the use of these two techno-pedagogical tools. Indeed, ISIS uses the Scratch programming environment, developed by researchers at the Media Lab of the Massachusetts Institute of Technology (MIT) for youth aged 8–16 [24, 25] with the goal, among others, of helping students explore various concepts in mathematics, science, and engineering design through creative, problem-solving, and collaborative activities—all of which are essential 21st-century skills [24, 26]. The fact that ISIS has been using contemporary pedagogical tools to facilitate programming learning for its future cohorts may indicate that it is aware of the importance of this discipline.

## 1.5 General research objective

ISIS has developed a relatively strong and well institutionalized educational system, in which school subjects have been reorganized (some eliminated, others added) along an integrative approach, with Salafi doctrine and action-oriented ideology (jihad) being central elements. The curriculum of all disciplines has been revised in line with these fundamental aspects, and a new body of textbooks has been developed along the same lines. It may come as a surprise to

some that ISIS decided to introduce a subject as modern and current as programming in its curriculum, and that the pedagogical objectives stated in the introduction to the textbook are in line with current curricular trends in technology education. Yet, a quick overview of the ISIS programming textbook also reveals it to be heavily focused on religious doctrine and military jihadist ideology.

The overall objective of this contribution to knowledge is to describe the programming curriculum of ISIS's educational system using the *Introduction to Programming with Scratch* textbook that has been produced, disseminated, and used by ISIS (including its pedagogical intentions, the curriculum, and indoctrination elements) from 2014 to 2017.

This study aims to shed light on the conception of education within ISIS. This study contributes to the constantly evolving corpus on curricula by shedding light on the peculiarities, methods, and objectives of the ISIS educational system, and that of learning-teaching technology in general. However, the results obtained are difficult to generalize, given the uniqueness of the educational material analyzed and the extreme context surrounding ISIS's educational reform. Thus, the interpretations only reflect the specific case of computer science education, specifically programming, under ISIS from 2014 to 2017. Nonetheless, these interpretations can be used in efforts to rebuild the educational systems in the affected regions and are relevant to the development of technology curricula. Beyond a simple descriptive intent, this study aims to provide a better understanding of ISIS's pedagogical intentions and approaches to contribute to the development of initiatives and interventions aimed at de-indoctrinating and demilitarizing the affected youth.

## 2. Theoretical framework

Before presenting the analyzed corpus and the research approach adopted, it is worthwhile to base our analysis on the theoretical frameworks surrounding curricula, textbooks, religious and military indoctrination, and programming learning.

### 2.1 Curriculum and indoctrination

There is no scientific consensus on the definition of *curriculum* [27–29]. According to Jonnaert [28], there are two major streams in curriculum studies. On the one hand, the Anglo-Saxon and North American approach to curriculum is systemic: it goes beyond the curriculum and takes into consideration the goals of the school system, teaching activities, assessment methods, and students' learning experience. On the other hand, the Franco-European approach sees the curriculum as a study program that integrates a certain orientation of school knowledge, as well as the structure or organization of school content and disciplines. We will adopt a definition inspired by Demeuse and Strauven [30] which lies between these two complementary approaches: the curriculum is a structured and coherent educational action plan, which articulates the aims of education and the pedagogical programming by drawing inspiration from the values promoted by society. In this regard, Jonnaert mentions:

> A curriculum is necessarily located downstream of educational policies and upstream of the daily teaching and learning actions in classrooms. As a result, a curriculum is in a way the interface between the broad orientations defined by educational policies and their implementation in classrooms.
>
> [28]

Curricula take shape when institutions make official choices regarding the contents and processes that must be taught in school. Curricula are adapted to local wills and needs and are

part of a global social perspective [28, 31, 32]. Under ISIS's education reform, for example, curricula and schooling more broadly participate in the process of "deep commitment," which fosters countercultural attitudes that promote the redefinition of history, values, and social struggles, and encourages members to engage in daily collective practices and rituals [11]. Thus, it is arguable that ISIS's curriculum and educational materials, such as textbooks, are part of a larger religious and military indoctrination operation [1].

## 2.2 Textbooks

Textbooks used in an institutionalized educational context are, in theory, developed following the official curriculum [31, 33]. Like education and curricula, they are embedded in a political context [34]. According to Wijaya et al. [35], textbooks play an even more significant role in education than curricula. Textbooks are both a pedagogical tool and a marketed product, both a teaching instrument and a learning object [33]. According to Mahmood [31], "Textbooks do not only influence what and how students learn, but also what and how teachers teach." Moreover, teachers rely heavily (and sometimes solely) on textbooks [32]. In most Arab countries, they are the primary resource for teaching and learning: "the textbook becomes the course outline, framework, and the parameters for the students' experience and testing, and worldview of science" [36]. This is especially true under ISIS, as teachers have no choice but to use official textbooks [4]. In this sense, the textbooks designed and disseminated by ISIS are a highly relevant corpus to better understand education within the regime. According to Olidort [1], ISIS's textbooks are instruments of a systematic indoctrination strategy, just like the curriculum in general, and participate in both a "Salafization" of Islam and an "ISization" of society. Thus, the pedagogical content and the dominant ideology (religious and military) are tightly intertwined in ISIS's textbooks [4].

## 2.3 Religious indoctrination

The primary meaning of "indoctrination" is pedagogical—literally meaning *instruction* or *teaching* [37]. Nowadays, this concept takes on a more complex, contextual, political, and all in all pejorative meaning. Indoctrination is now thought of as a form of teaching that prohibits knowledge of other perspectives, all of which are deemed false or even dangerous, and even goes beyond the idea of absolutism—belief in absolute principles, without nuance. The educational system becomes an instrument of indoctrination when reason gives way to faith or submission and when it teaches content that is subordinate to a religious or political state ideology (such as that of ISIS) and is no longer dissociated from it [37]. Educational authority is a powerful tool for indoctrination as schools provide access to a large segment of the population, which is ideal for expanding regime influence, and curriculum and textbooks can play a major role in the outcomes of a conflict, which is why totalitarian regimes place great importance on them [37, 38].

> Even when the educational system is totally subordinated to a political ideology and states its intention to radically change the didactic process and the training of individuals in order to make them obedient to the new political regime, the indoctrination intention is still hidden: the opponent is always the one who indoctrinates. The acquiring of new values and attitudes is not an (declared) action of indoctrination but has the role to "prepare" the young individual to "defend" himself and "fight" the enemy who is the enemy of the new political regime. This intention hides behind a necessity built according to the new axiological order.
>
> [37]

Totalitarian regimes thus instrumentalize education for political and ideological purposes via a grand campaign of indoctrination [37], within which teachers have a determining role in perpetuating social structures, anesthetizing creative power, and blinding students with irrationality. These regimes also tend to manipulate the curriculum (including sorting out disciplines, content, resources, etc.) to present only the positive aspects of a doctrine, even if it means falsifying content, and to inculcate hatred of anything that does not agree with their goals [39].

Under ISIS, the curriculum is strongly influenced by the dominant Salafi and jihadist doctrine. As the curriculum is at the center of educational reform [28], curricular resources and materials, such as textbooks, consequently bear the mark of this influence. ISIS's curricular model is thus centered on religious concepts, embedded in school subjects, which then serve as tools for Quranically guided education: "instructors are delivering the 'holy word' from their 'pulpits.' Their role is to transmit 'divine knowledge' to students, who must memorize and regurgitate it. [. . .] Finally, student exams are based on a system of rewards and punishments, both here on earth (e.g., prevention of marriage) and in the afterlife (e.g., promises of houris)" [13]. In keeping with ISIS's vision of an action-oriented religious doctrine (jihad), the curriculum is part of a larger religious and military indoctrination enterprise.

## 2.4 Military indoctrination

Militarism is defined by Naseem [38] as an "uncritical and unquestioning acceptance of the military by the general population": a normalization of conflict, violence, and war, mixed with patriotic and nationalistic values, all in the name of religious notions. An education that exposes students to these military and violent representations from a young age promotes desensitization to violence, but also militarization of society, particularly in young nations where preparation for war, supremacy of the military as an institution, and unquestioning loyalty to militaristic articulations of citizenship, nationalism, and patriotism are considered normal [12, 40].

For example, a study of the militarization of the learning subject in Pakistani textbooks [38] indicated that writing and discourse were used to lead learners to conceive of the "Other" as an ultimate enemy. Naseem [38] draws on feminist theories to analyze how the Other is constructed in highly militarized educational texts and finds that a binary and dichotomous view of the Other is prevalent. The Other is seen as the opposite of the self (the national self, the ideal citizen according to the dominant ideology): an evil, irrational, inferior, suspicious, and untrustworthy being against whom ideological and military conflict is inevitable. This construction of the Other is consistent with the mechanism of "covert indoctrination" via the educational system described by Momanu [37], where the Other becomes the one that indoctrinates and that must be fought and defended against.

The curriculum and textbooks are important tools of indoctrination under ISIS that are also instruments of militarization of society [1, 38]. ISIS militarizes textbook content namely by using violence-normalizing images, narratives, and symbols (e.g., ISIS flags, weapons of all kinds, tanks, missiles, representations of military training, etc.) and by tying learning elements to their usefulness in military training or jihad contexts [1].

Key to understanding this is not only the emphasis on violence or religious intolerance on their own terms but a very specific kind of program of indoctrination, ISization, in which both religious and general knowledge and skills are presented with the aim of training a fighter generation that will be able to wage attacks on behalf of the Islamic State's quartet of

interests—building an "Islamic" (Salafi) state, claiming a caliphate, using violent methods, and driving an apocalyptic narrative.

[1]

Since jihad is one of the pillars of Islamic State's ideology, it plays a clear and integral role in its curriculum, even though it is not one of the five obligations of Muslims (faith, prayer, fasting, charity, and pilgrimage to Mecca) [38]. This is yet another manifestation of the manipulation of the curriculum by ISIS for indoctrination purposes.

## 2.5 Overview of the scientific literature on ISIS textbooks

Two studies analyzing a corpus of textbooks used under ISIS, also part of the UNESCO Chair in Curriculum Development (UCCD) project on education in the Islamic state, have been published to date. The first consists of a content analysis of science textbooks for elementary school [2]; the second focuses on the integration of religious elements in textbooks (all subjects taken together) [13].

Potvin et al. [2] provide an analytical description of ISIS's pedagogical goals in science education through a content analysis of textbooks for students in all five grades of elementary school. The authors coded each sentence and image according to a mutually exclusive code system divided into five broad categories: (a) pedagogical objectives coded according to Bloom's taxonomy, to which religion-related, health and safety-related, social value-related, and jihad/war-related categories were added; (b) learning/informational content coded into emergent categories; (c) prescriptions or recommendations for teachers or students, coded according to their relationship to religion, health and safety practices, social values, or jihad/war; (d) pedagogical activities, coded according to Astolfi's et al. typology of elementary science activities; and (e) other content not relevant to the research objectives. The results show that the science curriculum presents mostly concepts, events, and phenomena, a relatively high proportion of which are scientifically erroneous, and that religious and health-safety considerations are more prevalent in the lower grades, while jihadist-warrior considerations are marginal and rather concentrated in the first and last grades (1st and 5th grade). The pedagogical objectives are overwhelmingly (90%) ranked at the lowest levels of Bloom's taxonomy, namely knowledge acquisition and comprehension, which calls into question how sophisticated and modern the curriculum is. Most of the learning content (60%) is simple and does not require a scientific approach to be observed, although as the level increases, more learning elements that require a scientific approach. Most of the content is presented in a dogmatic, decontextualized, or isolated manner. The content is often erroneous and inconsistent with scientific concepts and explanations. There are no explanations as to how the scientific knowledge presented has been arrived at and several religious elements are scattered throughout the textbook:

An uneven distribution of religious information and illustrations and their quasi-aesthetic function leads us to believe that the authors essentially conducted a cherry picking operation, inserting verses of the Qur'an into the documents to enhance themes for which keywords appear both in sacred texts and in typical school manuals [. . .] the rationale justifying their presence in the corpus is always rather thin, and could better be explained by an agenda other than scientific.

[2]

As for the prescriptions and recommendations, the authors hypothesize that the corpus has indoctrination ambitions since, even though the majority (56%) of prescriptions are health-safety related, a large number (26.7%) are religious. Regarding the proposed activities, the authors note that "the corpus examined essentially focuses on the more traditional 'transmission–memorization/repetition–application' approach rather than on developing scientific thinking through the 'functional–problem solving–structuring' triad" [2]. The authors argue that "the ISIS science education curriculum for the elementary levels can be viewed as 1) committed to an absolutist/totalitarian ideology, 2) inspired by a theocratic political program, and 3) scientifically and pedagogically poor" [2]. Scientific facts are presented indifferently from religious facts, and several dichotomies are present (good/evil, recommendable/unrecommendable, true/false, healthy/toxic, etc.). The corpus does not make a clear distinction between science and religion and Allah is often represented as the origin of scientific phenomena. Finally, the authors argue that the corpus in science is scientifically and pedagogically weak for several reasons: the pedagogical objectives are at the lowest levels of Bloom's taxonomy, the learning content is mostly common sense and declarative knowledge and does not encourage the development of critical thinking, several scientific facts are erroneous, no activity uses a scientific problem-solving approach, and several pedagogical objectives are inconsistent with the associated content and activities.

For their part, Arvisais and Guidère [13] explored and codified the integration of religious elements in ISIS textbooks using quantitative and qualitative content analysis. They calculated how frequently religious references were mentioned in the original textbooks in Arabic and analyzed the concepts, pedagogical objectives, and activities relating to religion (i.e., nature and origin of the element, position of the element in the lesson, function of the element in the pedagogical sequence). All religious elements identified throughout the textbooks (e.g., quotations from the Qur'an and the Prophet, concepts, themes, grammatical categories) originated from ISIS's religion textbooks, suggesting that a base of religious references must have been built up with the aim of integrating these elements into the textbooks of every discipline. The authors' analysis reveals that the worship of Allah is central to the educational approach of all disciplines. Learning the "religious sciences" is seen as an individual obligation for all believers within ISIS, while learning the "non-religious sciences" is seen as a collective obligation, i.e., some believers may engage in it while others engage in jihad. The analysis also showed that although the school and the mosque are both involved in education, textbooks explicitly specify that religious activities and calendars take precedence over educational activities and calendars. The curriculum is also divided between the Fundamental Programs (Doctrine, Quran, Tradition, Life of the Prophet, Calligraphy, Arabic, Islamic Education, and Physical Education) and Complementary Programs (science, mathematics, physics, chemistry, history, geography, computer science, and English), with the core curriculum (Fundamental Programs) being hierarchically superior. Religious elements are thus integrated into the textbooks of all disciplines: in the instructions provided to science teachers, who must discard the theory of evolution in favor of creationism; in the concepts and topics covered in history, which mix religious history with historical facts; in the prohibition of the use of traditional territorial markers or certain "non-Muslim" geohistorical concepts in geography; in the semiotic representation of Qur'anic, jihadist, or military elements in mathematics; or in the presentation of concepts in chemistry and physics in terms of their contribution to typically religious themes. The authors conclude by highlighting two key aspects of education under ISIS: it is centered on religious elements of the Salafi doctrine of Sunni Islam and is based on an action-oriented ideology. Thus, the curriculum is dogmatic in nature in all disciplines, which are presented as tools in the service of a religious doctrine strongly marked by jihad, and textbooks are major tools of indoctrination and radicalization. Finally, the authors argue that traditional education under

ISIS is first and foremost guided by the Quran, based on a system of reward and punishment, and does not allow for the development of critical thinking in students, but rather aims to develop young soldiers.

## 2.6 Learning programming in youth

In parallel to the previous sections of the theoretical framework, it is important to present some theoretical underpinnings of youth programming learning.

**2.6.1 Global historical perspective of programming curricula.** The idea of teaching computer programming to children in a school setting dates back to the 1960s [41] with the development of the LOGO language by Seymour Papert and his colleagues at MIT, a language created to teach programming as a skill and problem-solving tool. Papert later conceived of LOGO as both a coding environment and an environment for learning mathematics and developing procedural thinking [42]: "Programming a computer means nothing more or less than communicating with it in a language that both it and the human user can 'understand.' And language learning is one of the things that children do best" [41]. At the time, enthusiasm for programming spread rapidly, as it was credited with many virtues in the development of various cognitive skills, including planning and problem solving.

In the 1990s, interest in programming waned in favor of teaching new computer skills, such as word processing and Internet searching [43]. It was not until the new millennium that a renewed interest in coding emerged as part of the development of so-called 21st-century skills. The interest in these skills "is due, in large part, to numerous reports and whitepapers published in the last decade, and well-organized, well-funded education initiatives launched by non-governmental and not-for-profit organizations" [44]. The influence of these projects and the arrival of programming tools adapted to younger learners led several countries to reintroduce this learning into their elementary school curriculum, including Estonia, Greece, England, Australia, as well as some American states and Canadian provinces.

This return of programming in schools is attributable to three curricular intentions [45]. The first is to respond to an economic imperative that has become a leitmotif: to prepare the future workforce for the new technology industry. The second is to concretize the entrepreneurial ideal under which students are called upon to become "producers of innovations." The third is to master the basics of programming and to learn computational thinking, as these are considered essential for the development of 21st-century skills. These curricular intentions are relevant to one of the specific objectives of this article, which is to identify the pedagogical intentions of the ISIS curriculum.

**2.6.2 Scratch, a visual programming language to facilitate early learning in programming.** Since the introduction of computers in schools, the development of programming languages and software suitable for younger learners has facilitated the learning of programming [23], and the use of simulation software has become popular to contextualize and concretize abstract learning in this area [24].

More recently, MIT developed Scratch software as part of a research project to increase children's computer skills [24]. Scratch uses its own graphical programming language where the learner clicks and drags visual icons representing computer operations and functions such as loops, Boolean expressions, conditions, and variables to generate visual computer code, inspired by the idea of LEGO blocks. Thus, it is a set of "building blocks" meant to serve as a basis for students' creative expression, which is different from building something according to a specific pattern or goal: "This type of activity [following a specific pattern] might qualify as 'hands-on learning' or 'learning-by-doing,' but it is not what we mean by 'learning-through-designing.'" [26]. This programming environment eliminates syntax errors and increases the

accessibility of programming to young learners. In addition, Scratch has a simulation window that provides immediate visual feedback on the generated code in addition to contextualizing it by illustrating the purpose of the code. Scratch is complemented by a social network that allows for collaboration and project sharing. Students can create stories, games, music, animations, and much more: Scratch helps students think creatively, systematically, and work collaboratively, which are precisely the essential skills required for the 21st century [6]. The software's configuration and interface (color-coding; interlocking command, function, trigger, and control blocks; automatic display of coding errors; visual representation of each command's effect; grouping of related commands; independent coding for each object; etc.) promotes rapid and independent learning through observation and trial and error [25]. Scratch is free, available in 50 languages, and particularly popular in schools, as it is designed for easy and intuitive use by learners with no programming experience [25].

## 2.7 Specific research objectives

In connection with our general objective of describing the programming curriculum using the *Introduction to Programming with Scratch* textbook produced, disseminated, and used within ISIS, and considering the theoretical frameworks surrounding curricula, textbooks, religious and military indoctrination, and programming learning, as well as considering the contribution of previous studies on the matter, we can now identify the three specific objectives of this study:

1. Characterizing the curriculum's pedagogical intentions and definition of programming;

2. Characterizing the programming (computer science) curriculum;

3. Characterizing the religious and military indoctrination in the programming curriculum.

## 3. Methodology

In this paper, we will restrict our corpus to the *Introduction to Programming with Scratch* textbook published and distributed by ISIS and used as a teaching–learning tool from 2014 to 2017. After introducing the textbook, we will present the chosen research approaches for our specific research objectives.

## 3.1 Description of the analyzed textbook

The textbook used by ISIS was in circulation in northern Iraq between 2014 and 2017. Several hard copies were recovered by a local NGO from the ruins of Kirkuk in northern Iraq after its liberation in 2017. The corpus was completed and validated by an Internet search that allowed us to find these same manuals published online by ISIS itself, confirming that the analyzed corpus is reliable. In parallel, journalists and researchers associated with the New York Times have also recently obtained these documents [46]. All the textbooks were translated from Arabic to French by a professional translator with knowledge of Islam and the Quran.

The textbook *Muqaddimat al-barmaja bi-istikhdam sikratsh li-kafat sufuf al-marhala al-mutawassita* (*Introduction to Programming with Scratch*, hereafter referred to as *ISIS-Scratch*) [18] (Fig 1) is intended for 6th-grade students (the equivalent of the first year of high school) with no prior experience or knowledge of programming. The textbook is 148 pages long and, after an introduction to the ISIS curriculum and programming in general, covers the various commands for programming animations and games using Scratch software through a

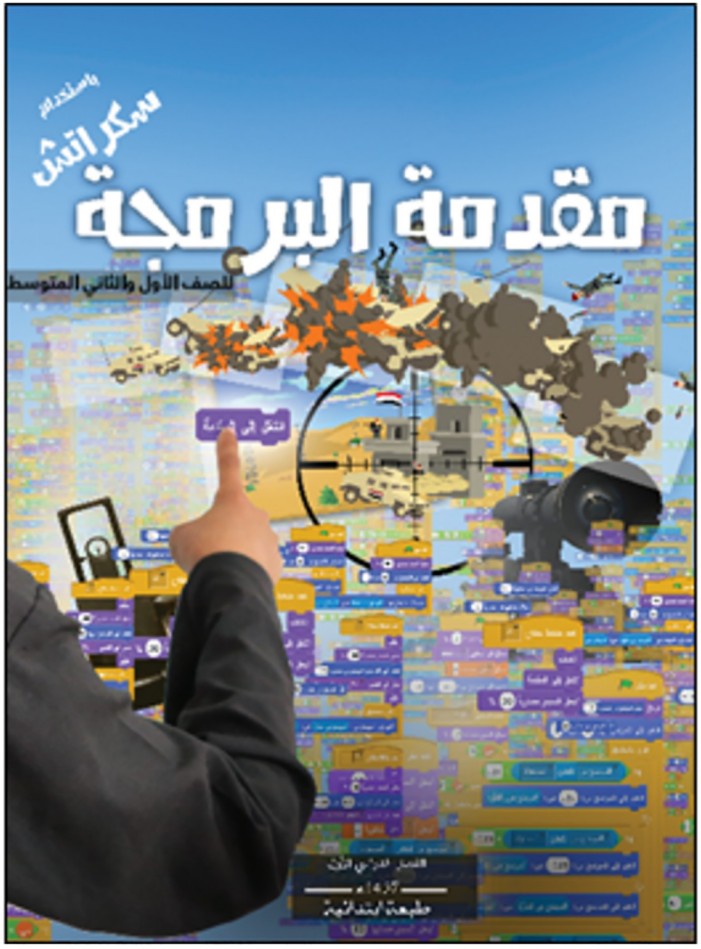

**Fig 1.** *ISIS-Scratch* **textbook cover.**

sequence of guided activities. The Results and Discussion section of this article provides further clarification of the manual.

## 3.2 Research approach

Some authors postulate that textbook research has three main orientations: the life cycle of the textbook from production to obsolescence (context); the reception of the textbook and its influence on different social groups (consumption); and product-oriented research, which examines the textbook more as a cultural and ideological vector and a pedagogical tool (content) [34, 47]. Other authors classify textbook analysis into three other categories or approaches: the horizontal approach, which focuses on the physical characteristics of the textbook and its organization; the vertical approach, which analyzes the presentation and treatment of content; and the contextual approach, which examines the use of textbooks in school settings. Furthermore, according to Pingel [32], when analyzing textbooks, "text analysis should be complemented by an examination of illustrations, as visual elements are becoming more prominent in today's textbooks and tend to appeal to readers' emotions to create lasting impressions." The present study adopts a product-oriented type of orientation—which sees

textbooks as cultural and ideological vectors—and a vertical approach—where textual and iconographic contents are at the heart of the analysis.

The chosen method of analysis is content analysis, a grounded research technique for an objective, systematic, and quantitative description of the manifest and latent content of communication [48]. This method identifies and codes the salient trends and themes of the corpus under study according to categories chosen based on a general objective [49]. Thus, it is a mixed approach aimed at quantifying and qualifying content according to research objectives [32].

## 3.3 Data analysis

The present content analysis is done in three parts, corresponding to our three specific objectives: a discourse analysis; an analysis of the quality of the pedagogical sequence; and an iconographic analysis. These analysis have been conducted manually.

The first step aims to 1) characterize the pedagogical intentions through a discourse analysis to identify what kinds of information and organization are valued in the textbook [32] and identify the curricular intentions. To do this, each section of the textbook, constituting a unit of discourse analysis of some type (introductory material, programming rudiment, challenge, exercise, lesson, etc.), is summarized and analyzed.

The second step aims to 2) characterize the programming curriculum. According to Winch's [50] work on curriculum design, it is essential for a curriculum to introduce the three major types of knowledge: concepts, propositions, and know-how. In order to identify these within the ISIS curriculum, we rely on Bloom's revised cognitive domain taxonomy [51] adapted to computer science learning [52] to analyze the quality and relevance of programming pedagogical sequences. According to this taxonomy, the pedagogical objectives (and their associated action verbs) are divided into six hierarchical levels (from the simplest learning, at level 1, to the most complex, at level 6), as described below. For each level, an example of a pedagogical objective in programming (task example) is given. Quality pedagogical sequences that are effective for student learning target pedagogical objectives at a variety of levels and ideally include high-level cognitive operations [51].

- Level 1) Knowledge: To memorize, name, or reproduce information in terms similar to those learned; to locate information; to know and state or list events, dates, places, facts, big ideas, rules, laws, formulas. Task example: Naming the elements to be defined for each programming object.

- Level 2) Comprehension: To understand, translate, and interpret information based on what has been learned; to grasp meanings; to translate or generalize knowledge; to interpret or explain facts from a given framework. Task example: Identifying the programming block that makes an object move.

- Level 3) Application: To select and transfer data to perform a new task or solve a problem; to reinvest methods, concepts, and theories in new situations; to solve problems using the required skills and knowledge. Task example: Changing the controls so that an object moves three times instead of two before disappearing.

- Level 4) Analysis: To distinguish, classify, and relate facts and structure of a statement or question; to perceive patterns; to distinguish implied meanings; to extract elements; to identify constituent parts of a whole to distinguish ideas. Task example: Explaining the steps involved in programming a tennis game where the score is displayed on the screen.

- Level 5) Synthesis: To conceive, integrate and combine ideas into a proposal, a plan, a new product; to use available ideas to create new ones; to generalize from several facts; to relate knowledge from several fields. Task example: Designing a shooting game on Scratch.

- Level 6) Evaluation: To estimate, evaluate or criticize according to self-constructed standards and criteria; to compare ideas; to determine the value of theories and presentations; to make choices based on reasoned arguments; to check the strength of evidence; to judge the degree of subjectivity. Task example: Deducing the software controls used by observing an electronic game.

In addition, we will focus on the learning progression [50]. To do so, we will identify the basic concepts and skills in programming as identified by Tew and Guzdial's [53] conceptual inventory of programming for novice learners, namely: sequencing, logical operators, state selection and conditions, definite and indefinite loops, arrays, control functions, value return functions, variables, recursion, and reading, writing and debugging code. We will then identify the main pedagogical strategy used.

Finally, the third step aims at 3) identifying the elements of military and religious indoctrination by coding the context surrounding each pedagogical activity (discourse analysis) and each image in the textbook (through an iconographic analysis) according to its military, religious, or other nature. Iconographic analysis is a method of interpreting visual content, "a merely descriptive method, aiming at an objective and neutral description and classification of depicted motifs" [54]. To do so, we consider all foreground (Scratch objects and images used as decoration on the page) and background images as units of analysis (a repeated image is counted only once), excluding screenshots of the Scratch software interface or control blocks. Each unit is then coded according to whether it is military, religious, or other; for the "other" category, the theme is then indicated (e.g., animals, characters, scenery element, etc.). This coding will allow us to present a percentage analysis of the context and images of the activities in the textbook.

The following figure (Fig 2) aims to summarize the categories and codes used in the content analysis.

## 4. Results and discussion

The following sections present the *ISIS-Scratch* textbook analysis results and related discussion within the limits of our chosen theoretical frameworks.

### 4.1 Pedagogical intentions of the curriculum

The *Muqaddimat al-barmaja bi-istikhdam sikratsh li-kafat sufuf al-marhala al-mutawassita* (*Introduction to Programming with Scratch* or *ISIS-Scratch*) textbook begins with a general introduction common to all textbooks and presents the educational intentions of the ISIS curriculum, which is to align learning with the Qur'anic approach, and more importantly, to understand and integrate the vision of the early companions of the Prophet, as well as to establish the Islamic Caliphate. This apologetic text begins as follows:

In the name of Allah the Most Gracious, the Most Merciful, praise be to Allah, honoring the believers with His victory, lowering the unbelievers by His power, executing all action by His will, dragging down the disbelievers by His judgment, the One who determined the alternation of days by His righteousness and made the end happy for the pious by His grace, and may Allah's peace and prayer (blessing) be upon him who has erected the beacon of Islam by his sword. By the grace and help of Allah the Almighty, the Islamic State is now entering a

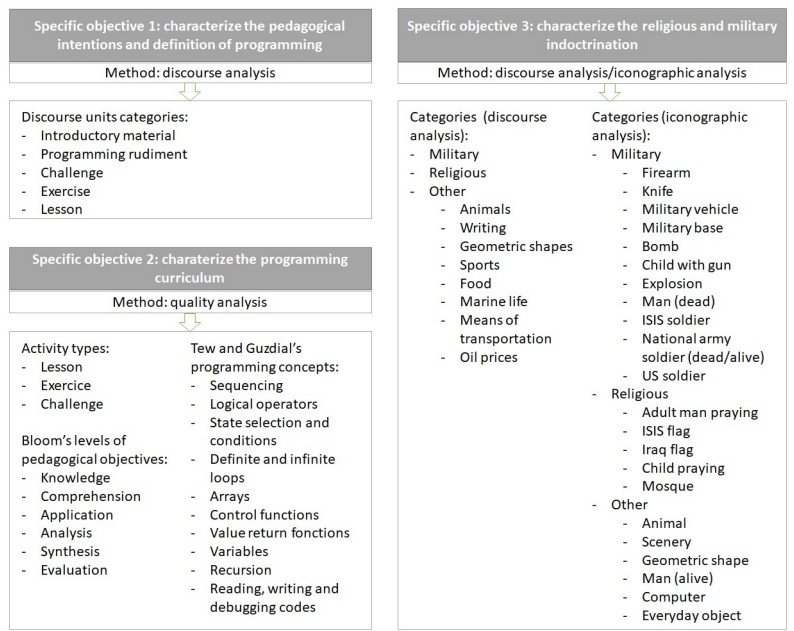

**Fig 2. Content analysis categories and codes.**

new era by laying the foundation stone of Islamic education based on the method [approach] of the Qur'an and the Prophetic guidance and based on the understanding of the way of the venerable ancestors and the first pious predecessors [of the Prophet] and by a clear vision, neither Eastern nor Western, but Qur'anic prophetic that is far from the passions, heresies and impostures emanating from the missionaries of Eastern socialism and Western capitalism or the agents of vicious parties and movements in the four corners of the world.

[18]

The rest of the introduction praises ISIS for being responsible for putting Muslims back on the right track and for building the caliphate.

Next, a specific introduction to the textbook presents the intentions of the programming curriculum. The intentions are to train a generation of mujahideen capable of using and developing high technology, the core element of modern weaponry, and to train scientific cadres capable of transferring this core of Western civilization to the Muslim East to establish a digital platform for ISIS. The introduction then lists the reasons for choosing the Scratch software for teaching and learning programming and, finally, points out that learning programming at a young age is a key step in achieving technical and scientific autonomy for ISIS.

Why learn programming? Today, our Islamic State is under attack by all kinds of modern weapons from the military coalition of crusader infidels. It is through jihad that we fight back, and it is through Allah's blessing that our state will survive and expand despite the hostility of the ungodly, Arab, and Western countries. The Islamic State needs creative programmers who are capable of developing and upgrading its weaponry system, in accordance with Allah's command: "And prepare [to fight] against them all you can as a force and as equipped cavalry, in order to frighten the enemy of God and your own," programming being an important part of modern weaponry.

[18]

ISIS states that its mission is to prepare and qualify students to use modern and current technologies and to develop their programming skills, independence, and creativity in science and technology. It should be noted, however, that the ISIS programming curriculum does not appear to be part of a "mathematical" approach focused on problem solving—a 21st-century skill with which programming learning is often associated [55]. Instead, this curriculum takes a more superficial "utilitarian" approach that focuses on completing simple step-by-step tasks to have students learn how to use specific software rather than having them understanding how it works.

Thus, the training of a "generation of mujahideen who will know how to use and also develop these modern technologies in times of peace and war" [18] referred to in the introduction to the *ISIS-Scratch* textbook and the utilitarian approach used throughout the textbook suggest that the intentions of ISIS's programming curriculum stem from military and religious imperatives much more than from a stated desire to foster 21st-century skills. Furthermore, despite some introductory passages in the textbook referring to international trends in teaching–learning programming, ISIS's intentions remain very clear. This is consistent with the definition of *curriculum* stated earlier, according to which the curriculum is a structured and coherent plan of educational activities that articulates the goals of education and the pedagogical programming by drawing inspiration from the values promoted by society [30]. In this case, it is clear that ISIS's programming curriculum is based on its wishes and needs relating to jihad and the building of the caliphate. As Arvisais and Guidère [4] (authors' work) aptly mention, the educational content and the dominant ideology of ISIS are intertwined to such an extent that it is impossible to consider them separately.

Following the general introduction and the textbook-specific introduction, examples are offered to illustrate how programming can be used. The textbook states that programming involves instructing electronic systems to perform different tasks: ringing a phone, setting a car alarm, turning on an air conditioner, or bombing missiles from enemy aircraft [18]. An operational definition of *programming* is then presented as being "the process of writing instructions and controlling a computer or other device (electronic or including electronic equipment) to direct and inform it how to process data and how to perform a series of required actions" [18]. We consider this definition to be fair, adequately popularized, and consistent with definitions often presented in the field, for example: "computer programming is defined as the process of developing and implementing various sets of instructions to enable a computer to perform a certain task, solve problems, and provide human interactivity" [56].

Then, the textbook gives reasons for learning programming, which mainly resemble jihadist injunctions from ISIS. In contrast to the curricular intentions usually associated with youth learning programming, which we presented earlier (economic imperative of workforce training; entrepreneurial ideal of innovation; development of computational thinking) [45], we consider ISIS's curricular intentions to rather be oriented toward indoctrination, militarism, and preparation of a "war workforce," as evidenced by the following excerpt: "The Islamic State needs creative programmers who are capable of developing and upgrading its weapons system, in accordance with Allah's command" [18].

The Scratch programming language is then defined as a language—like Arabic, Russian, and English—meant for communication with electronic systems. The Scratch software is then presented. The version used by ISIS is a revised version adapted to its specific curriculum [18]. Regarding programming basics [18], the textbook outlines what needs to be determined for each programming object (class, behaviors, features, and specifications), how they are designed and used in Scratch, as well as the main components of the software interface and examples of programming blocks. For example, on page 16 (Fig 3), the textbook explains how

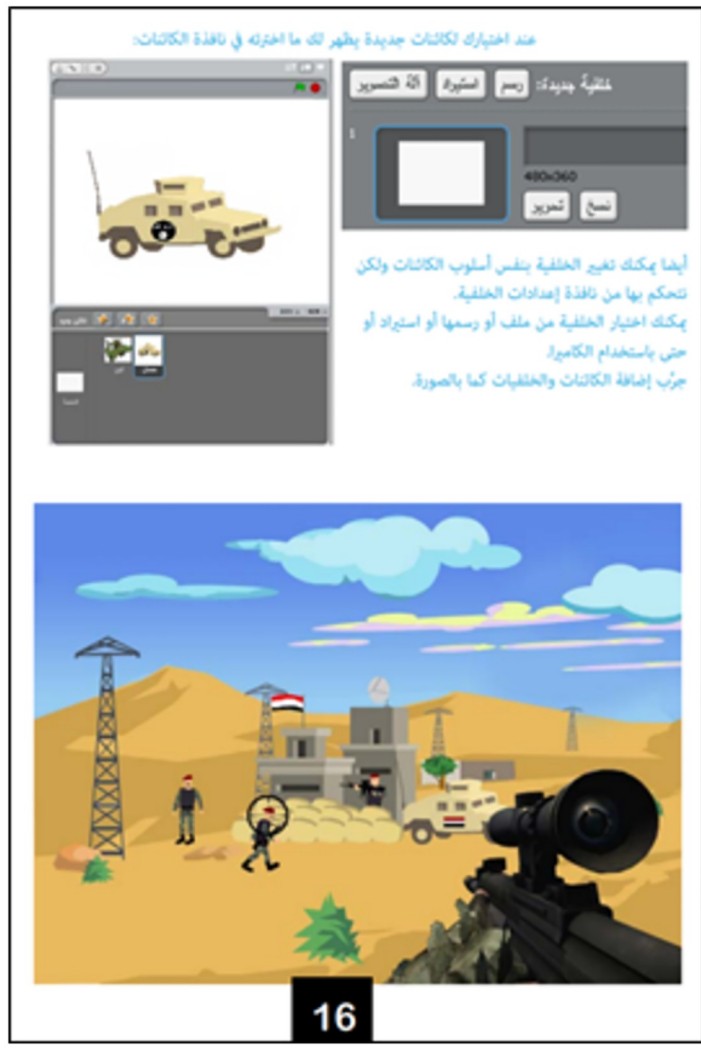

**Fig 3. How to integrate objects and import a background** [18].

to integrate one or more programming objects and how to import a background on which to place these objects.

The rest of the textbook [18] is a sequence of various activities and will be discussed in the next section, which addresses our second specific objective.

Religion and militarism are strongly emphasized in the many introductory pages, in many of the proposed learning activities, and in the iconography throughout the textbook, which blur the lines between promoting programming as a modern skill to be taught and using programming as a jihad tool for building ISIS caliphate. For example, the "Horse Challenge" involves programming a game in which "fighters will go through infiltrating the apostates' army [Iraqi/Syrian soldiers] to reach the explosives and use them to fight their way to one of the infidels' barracks and forts, with the goal of controlling and occupying them" [18]. Here, the teaching of programming is mixed with the teaching of a military strategy. The military and religious indoctrination intentions of ISIS's programming curriculum are evident from the introduction and are thus repeated throughout the textbook.

## 4.2 Programming curriculum

The rest of the manual consists of a sequence of activities on the basic programming concepts, the study of which allows us to better characterize the ISIS curriculum. Since textbooks are developed from the official curriculum and, in many countries, are the only teaching–learning medium [31–33, 36]—especially under ISIS where their use is mandatory—it can therefore be considered that in this particular case, and in the current state of knowledge about ISIS's educational system, the textbook of a discipline is a faithful representation of that discipline's curriculum.

This curriculum takes the form of a textbook including a pedagogical sequence in paper format that puts the student in action with the visual programming software Scratch. This sequence includes 21 activities of three types: lessons, exercises, and challenges (or a combination of two of these three types). A *lesson* is an activity where the student only memorizes information (knowledge, 1st level of Bloom); the student does not interact with Scratch software. An *exercise* is an activity where the student interacts with Scratch software. It should be noted, however, that the exercises in the textbook are very structured, so that the student must simply replicate a code presented to them in the textbook; they are not required to compose their own code. According to Bloom's adapted taxonomy presented in the previous section, these exercises therefore only work on the student's understanding (Bloom's 2nd level) of basic programming concepts. Finally, at the end of some of the exercises, an additional *challenge* is sometimes proposed to the student. The challenge requires the student to apply new knowledge in a different context (application, 3rd level of Bloom). It is important to note that some activities have been identified here as exercises, even though they are titled "challenges" in the textbook (literal translation), in accordance with these definitions of activity types.

Table 1 presents a summary of the 21 activities and their context, associated with a type of activity (lesson, exercise, or challenge) and with one level of Bloom's taxonomy. Each activity is also associated with the programming learning that it addresses.

Five of the activities (23.8%) are of the lesson type. This small proportion of lessons seems to indicate a desire to engage students in concrete action through exercises or challenges, with nine out of 21 activities (42.9%) being of the exercise type. As mentioned earlier, these exercises aim to reproduce lines of coding presented in the *ISIS-Scratch* software textbook. Instructions are highly explicit, and students must simply follow a pre-established step-by-step approach to complete the exercises. Finally, seven out of 21 activities (33.3%) incorporate an additional challenge. These challenges represent rare opportunities for students to apply what they have learned to another context independently, that is, without the textbook providing the solution. These challenges are the only activities where students are asked to produce original code. It should be noted that none of the activities proposed deal with reading or debugging code, which are two important programming skills.

Considering that the pedagogical objective of the lessons is memorization and that the exercises aim at developing an understanding of the programming logic through the reproduction of code, two-thirds of the curriculum (66.6%) achieve the first two levels of Bloom's taxonomy (memorizing and understanding). Only the challenges (33.3%) aim at application, which is the 3rd level of Bloom's taxonomy.

In terms of learning progression, several of the basic programming concepts identified by Tew and Guzdial [53] are present, and the ordering of these appears consistent with a classical ordering, such as that proposed by code.org. Code.org is a website that hosts visual programming courses which is particularly interesting to younger children since the content is adjusted and presented using instructional scaffolding adapted to their prior knowledge [57, 58]. Therefore, the progression used by ISIS is very similar to the one used by code.org. For example, the

**Table 1. Pedagogical Sequence of the ISIS programming curriculum.**

| Activity | Activity title [and context] | Activity type | Bloom level | Learning progression |
|---|---|---|---|---|
| 1 | Introduction [Context: learning programming is useful for jihad and building the caliphate] | Lesson | (1) Knowledge | Definition or programming and curricular intention |
| 2 | Programming blocks [No context: student forms a compilation of building blocks] | Lesson | (1) Knowledge | Visual programming and interface |
| 3 | Animal Sounds Challenge [Context: playing the sounds associated with different animals] | Exercise and challenge | (2) Comprehension and (3) application | First lines of code to play sounds |
| 4 | Circle Color Change Challenge [Context: randomly changing the color of a circle] | Lesson | (1) Knowledge | Adding images to a project |
| 5 | Learning to pray [Context: showing prayer positions, movements, and words] | Exercise | (2) Comprehension | Recording sound and sequencing multiple sounds |
| 6 | Moving the car [No context: student moves objects] | Exercise | (2) Comprehension | Moving an object |
| 7 | How to save the project [No context: student saves a project] | Lesson | (1) Knowledge | Saving a project |
| 8 | Kalashnikov Challenge [Context: simulating fighting with a Kalashnikov] | Exercise | (2) Comprehension | Adding images to a project |
| 9 | Repeat command [No context: student adds a definite or infinite repeat] | Lesson | (1) Knowledge | Concept of definite and infinite loop |
| 10 | Shoot Down Helicopter Challenge [Context: shooting down an enemy helicopter flying over ISIS airspace with a firearm] | Exercise and challenge | (2) Comprehension and (3) application | Infinite loop stopping condition |
| 11 | Bat [Context: making a bat fly] | Exercise | (2) Comprehension | Infinite loop stopping condition |
| 12 | Geometric shapes [Context: drawing geometric shapes] | Exercise and challenge | (2) Comprehension and (3) application | Moving objects with definite loops |
| 13 | Programming more than one object [No context: student programs multiple objects] | Exercise | (2) Comprehension | Moving objects with definite loops |
| 14 | Marine life [Context: representing marine life by animating marine organisms] | Exercise | (2) Comprehension | Moving objects with definite loops |
| 15 | Number Challenge [Context: writing a number after it is displayed and named] | Exercise and challenge | (2) Comprehension and (3) application | Variables and Cartesian quadrants |
| 16 | City Life Challenge [Context: helping a cat get through a city by overcoming several challenges] | Exercise and challenge | (2) Comprehension and (3) application | Conditional loop |
| 17 | Tennis Challenge [Context: creating a tennis game where the player plays against the computer] | Exercise and challenge | (2) Comprehension and (3) application | Incrementing and displaying variables |
| 18 | Apostate Hunt Challenge [Context: killing apostates with a gun operated by a sniper] | Exercise and challenge | (2) Comprehension and (3) application | Logical operators |
| 19 | Farm Challenge [Context: picking apples that fall from the tree into a basket] | Exercise | (2) Comprehension | Integrative exercise |
| 20 | Horse Challenge [Context: infiltrating and bombing an enemy military base to occupy it] | Exercise | (2) Comprehension | Integrative exercise |
| 21 | Dinar Challenge [Context: identifying the dinar that weighs less than the others; the Dinar is used because ISIS refuses the US dollar] | Exercise | (2) Comprehension | Integrative exercise |

concept of sequencing, which is the basis for subsequent learning, is addressed early in the pedagogical sequence. As for the concept of loop, it is first introduced, and then deepened by adding the concept of conditional loop. It appears, however, that logical operators should have been introduced before conditional loops since these loops require the use of a logical operator. This represents a limit to the proposed learning progression. We also note that some concepts (e.g., variables and their incrementation) are covered too briefly, considering their importance for the integration activities proposed at the end of the pedagogical sequence. Despite these limitations, the learning progression shows that ISIS has clearly thought through the didactics of programming.

As for the contextual content of the activities, 23.81% of the activities do not include any contextual passages and are introduced by a simple presentation of the knowledge to be

acquired, such as in this example from the "Programming more than one object" exercise: "In the previous lessons, we often programmed one object, in this lesson we will learn to program more than one object." [18]. The remaining activities (76.19%) relate to contextualized real-life elements, for example the "Animal Sounds Challenge": "In this example, we will recognize some animal sounds" [18]. In addition, there is no common thread between the activities in terms of their contexts: they range from "ordinary" or childlike contexts (involving animals, shapes, and colors) to religious or military and violent contexts. The military or religious indoctrination fostered by the contexts surrounding the learning activities is discussed in more detail in the next section, dealing with our third research objective.

This textbook being the only ISIS textbook to address programming and, by extension, technology or digital technology, it, therefore, represents the entirety of the ISIS computer science curriculum. As such, it is relevant to compare it to the programming curriculum of other organizations, such as the Computer Science Teachers Association (CSTA) Curriculum standards (USA), the Royal Society Report (UK), and the Joint Informatics Europe & ACM Europe Working Group (Europe-wide affiliation and focus). These organizations were selected and compared because they are recognized bodies in computer science education for youth [59]. To operationalize this comparison, we highlight the four key elements identified by Webb et al. [59] in their comprehensive curriculum analysis, namely overview of discipline, key areas of conceptual knowledge, techniques and methods, and ways of thinking. Table 2 compares the different curricula for each of the key elements along with the ISIS curriculum.

As can be understood from the table, several differences exist between the ISIS curriculum and those of other organizations. First, ISIS seems to see this discipline as an opportunity for military and religious indoctrination, unlike the other curricula which remain unbiased in terms of defining the discipline and its usefulness. In addition, the ISIS curriculum focuses solely on programming, whereas other curricula have a broader focus and incorporate skill development and societal issues into the very definition of the discipline.

In the ISIS curriculum, the teaching techniques and methods, because of their very restrictive nature, focus on the reproduction of codes rather than on developing students' coding and problem-solving skills, which is what other curricula do, particularly through more open-ended situational exercises. Under ISIS, the teaching emphasizes memorization and reproduction. This type of guidance also suggests that teaching is lecture-based. In this sense, ISIS seems to be aware of the potential of learning programming among young people (i.e., the "useful" aspect of programming that can influence the capacity of ISIS in terms of recruitment, propaganda, and weaponry), but seems to have given little thought to the pedagogical and didactic aspects specific to this discipline. There is also little room for creativity: students must use pre-produced visual elements (Scratch objects) and program them following a fully defined procedure to achieve a predetermined result. Thus, ISIS does not seem to converge with the digital teaching methods suggested as most effective [60], which the other curricula presented do.

From these teaching methods come limited learning opportunities in comparison to international curricula. These learnings only fit into levels 1–3 of Bloom's taxonomy (mainly levels 1 and 2), without approaching the higher levels (4, 5, and 6) that allow the development of complex skills. The development of programming skills, and by extension, the development of computational thinking, therefore, does not seem to be part of the ISIS curriculum's aims, unlike other curricula. It is nevertheless possible that the learning suggested by the ISIS curriculum may contribute to the development of logical thinking, as defined by Yunus [61], since it addresses the use of logical operators and their sequential integration within lines of code. Nevertheless, although several exercises aim at reproducing lines of code including logical operators, few activities effectively allow the application of these skills.

**Table 2. Comparison of youth programming curricula.**

| | Computer Science Teachers Association (CSTA) Curriculum standards (USA) | Royal Society Report (UK) | Joint Informatics Europe & ACM Europe Working Group (Europe-wide affiliation and focus) | ISIS |
|---|---|---|---|---|
| **Overview of discipline** | The study of computers and algorithmic processes, including their principles, their hardware and software designs, their applications, and their impact on society. | Encompasses foundational principles, widely applicable ideas, and concepts as well as techniques and methods for solving problems and advancing knowledge as well as a distinct way of thinking and working. | Informatics is a distinct science, characterized by its own concepts, methods, body of knowledge, and open issues. | Programming is the process of writing instructions and controlling a computer or any other [electronic] device and is used to establish a digital platform for ISIS, and to train scientific cadres to transfer the center of gravity of universal civilization from the unbelieving West to the Muslim East. |
| **Key areas of conceptual knowledge** | Computational thinking; collaboration; computing practice; computers and communication devices; and community, global, and ethical impacts. | Programs; algorithms; data structures; architecture; and communication. | Algorithm; performance and complexity; data structure; concurrency (parallelism) and distribution; language (including programming languages); abstraction. | Programs in visual programming. Knowledge about coding and mathematics (variables and simple operators). |
| **Techniques and methods** | A wide range of techniques and methods for: analyzing massive data, solving problems, authentication, recursion, object-oriented operations, modeling, searching, testing. | Methods or ways of thinking from computer sciences, including: Modelling—representing chosen aspects of a real-world situation in a computer. Decomposing problems into sub-problems, and decomposing data into its components. Generalizing particular cases of algorithm or data into a more general-purpose, re-useable version. Designing, writing, testing, explaining, and debugging programs. | Problem-solving techniques include: Representing information through abstractions. Logically structuring and analyzing data. Automating solutions through algorithmic thinking. Identifying, analyzing, and implementing possible solutions with goal of achieving the most efficient solution. Formulating problems in a way that facilitates the use of a computer. | Memorization of knowledge about programming through lessons; understanding of programming concepts through replication of lines of code; application to other contexts through challenges. |
| **Ways of thinking** | Algorithmic thinking; computational thinking; scientific thinking; logical thinking; critical thinking. | Computational thinking; logical thinking; systematic thinking; analytical thinking. | Confidence in dealing with complexity. Persistence in working with difficult problems. Tolerance for ambiguity. Ability to deal with open-ended problems. | *Logical thinking*, replicating a specific sequence. |

Overall, this analysis is in line with Arvisais' and Guidère's observations about the entire ISIS curriculum [13] (authors' work): "[. . .] the ISIS teaching methodology seems to be downright traditional and stems from the premise that instructors are delivering the "holy word" from their "pulpits." Their role is to transmit "divine knowledge" to students, who must memorize and regurgitate it."

## 4.3 Military and religious indoctrination elements

Most learning activities (76.19%) in the *ISIS-Scratch* textbook have a contextual introduction relating to real-life situations, which we have categorized as military, religious, or other. Of these exercises, challenges, and lessons, four (25%) are prefaced with military-oriented contextual introductions focusing on the importance of slaughtering the enemies of Allah and ISIS. Of these four activities, three include a Qur'anic surah that serves as a justification for the context. For example, the "Kalashnikov Challenge" is introduced as such:

"And prepare [to fight] against them all that you can as a force and as equipped cavalry, that you may frighten the enemy of Allah and your own and others whom you do not know

besides these, but whom Allah knows. And whatever you spend in the way of Allah will be amply repaid to you and you will not be harmed in any way."

Surah The Spoils: 60

Allah has commanded us to prepare all kinds of forces, especially military force, to fight the enemies of Allah and our enemies. The Kalashnikov rifle is one of the most commonly used weapons by Islamic State soldiers. In this challenge, students will learn about this weapon and how to program Scratch to simulate a fight with a Kalashnikov, which allows them to see the bullet come out, the sound it makes, and also its shape.

[18]

The first paragraph here is a Qur'anic surah that serves as a justification for the second paragraph, which is the context of the exercise.

Two activities (12.5% of the contextualized activities) are religious in nature. First, "Learning to Pray" (Fig 4), which explicitly aims to "teach children to pray in the right way" [18], also includes a quotation from a Qur'anic surah. The "City Life Challenge," which involves

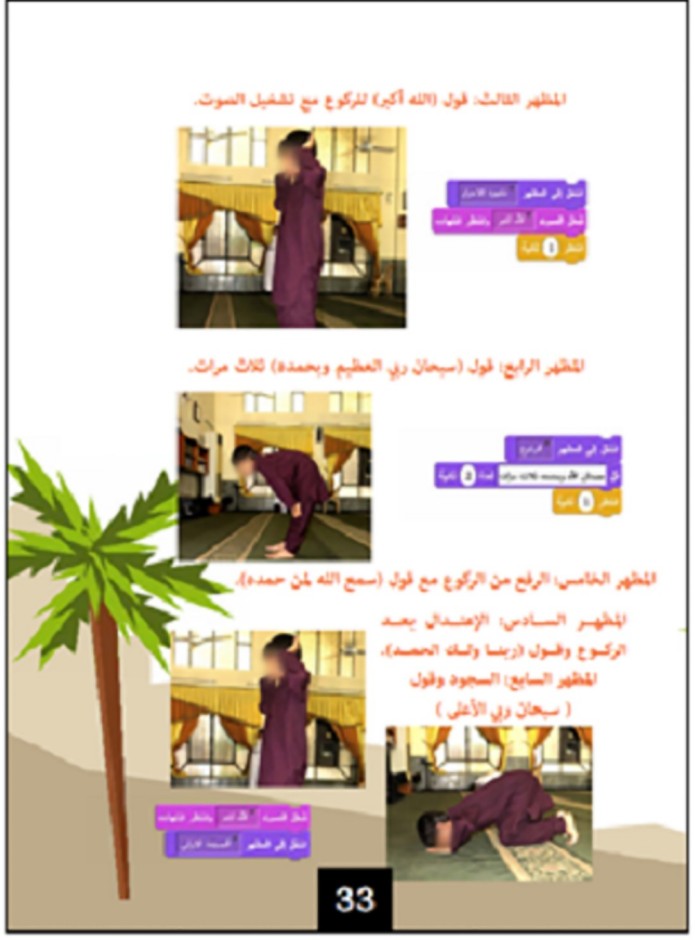

**Fig 4. "Learning to pray" exercise [18].**

programming a game in which a cat must travel through a city avoiding various obstacles, is introduced with an Islamic law, which commands "to be merciless to disbelievers and hypocrites and resist them in the most violent ways [but] to have compassion for believers and vulnerable beings, even non-human ones, whom Allah has created on earth and in the heavens" [18].

The remaining contextualized activities (62.5%) are non-military or non-religious in nature. The themes are: animals, writing, geometric shapes, sports, food, marine life, means of transportation, and oil prices. While the military and religious contextual passages are all longer than a paragraph, the other contextual passages are very short (a few sentences). For example, the "Tennis Challenge" is introduced simply with the following: "In this stage, we will face a professional tennis player, which is the computer. We will create a tennis court and play against the opponent. Note that each player has 5 points" [18].

The very last activity in the textbook deserves special attention. The "Dinar Challenge" [18] (Figs 5–7) is not strictly speaking military or religious in nature (its theme is "oil prices"), but its context is certainly highly politicized, and the idea of reinstating the dinar is highly linked to the totalitarianism of ISIS, in that it refers to the glorification of the past caliphates. This idea can be considered religious in nature, as caliphates are theocratic political structures. Moreover, the iconography associated with this activity is mostly military in nature and unrelated to the programming skills needed. While the activity is a logic problem in which the student must program a game to find among four identical objects (dinars) the one that weighs less than the others, the contextual introduction of the exercise speaks instead of "the greatest swindle, theft, and fraud operation in history carried out for decades until today" that is the sale of oil at a price that is too low by the "states of the tyrannical crusading coalition"—the United States [18].

The activity is contextualized in the following way: ISIS decrees that oil is no longer tied to the US dollar because the miscreant state (the US) was forcing the rest of the world to sell oil at a price that does not reflect its true value, which is "the greatest swindle in history." An image

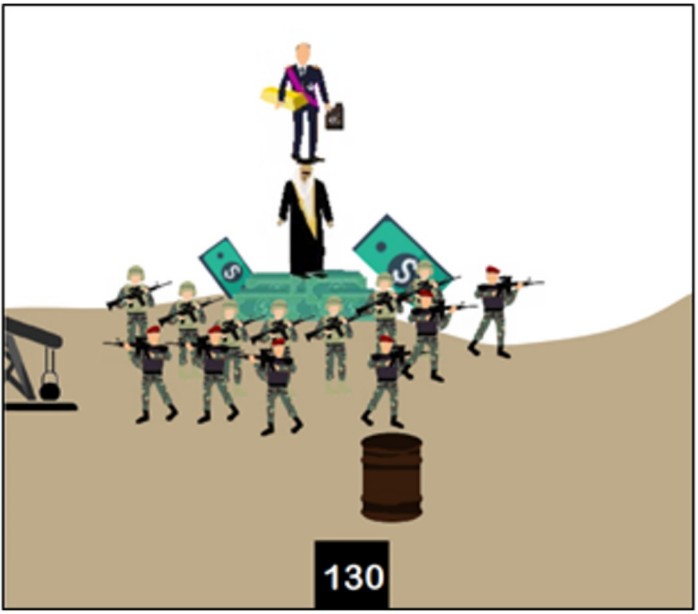

**Fig 5. "Dinar Challenge" 1 [18].**

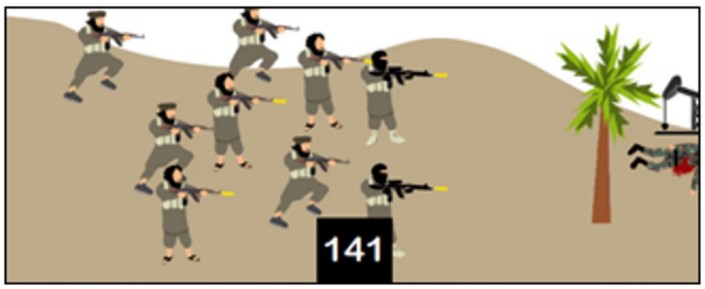

**Fig 6. "Dinar Challenge" 2 [18].**

on p. 130 (Fig 5) depicts American (or Western) soldiers protecting dollar bills on which a man is standing whose attire appears to represent the rulers of the Gulf States (an Arab king). On the head of this man stands a visibly Western man, probably American, who holds a gold bar in one hand and an oil can in the other. In the activity, the currency used is the gold dinar, which was in circulation during the golden age of Arab-Muslim civilization (under the Umayyad and then Abbasid caliphate). The narrative represented by the iconography of this challenge continues with images of oil wells protected by American and Iraqi/Syrian soldiers, followed by the arrival of armed ISIS soldiers [18] (Fig 6) shooting at them. The non-ISIS soldiers are depicted in a very graphic way (shot, lying on the ground, one with blood flowing from his abdomen, and others whose heads are reduced to pieces of brains in a pool of blood).

On the last page of this activity [18], the power relationship explained at the beginning of the challenge is reversed (Fig 7). Two Islamic State soldiers are shown: a bearded man holding a gun with a turban on his head, and another blond man, also bearded, holding a wireless

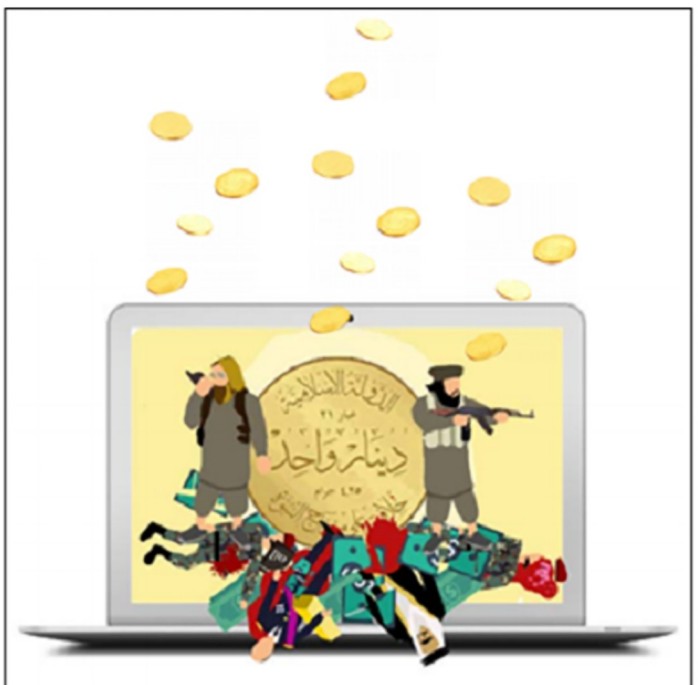

**Fig 7. "Dinar Challenge" 3 [18].**

communication device. The two men stand over dollar bills, Iraqi/Syrian soldiers of the national army, the Arab king (representing the Gulf countries), and the Western man—all shot and bleeding. In the background is a gold coin, the dinar, on which is written: "The Islamic State, 21-carat gold, a caliphate according to the prophetic path." This violent narrative is justified by the contextual introduction of the activity, in which the United States is represented as the enemy—the con artist, the miscreant—to be defeated and as the "Other" against whom combat is inevitable and violence is justified [38].

Thus, the *ISIS-Scratch* textbook contributes to the normalization of violence and war in the name of religious or political ideas [38] and, by exposing young people to violent military representations—often leaving very little room for imagination—promotes desensitization to violence [12, 40]. On this point, another image in the textbook is particularly telling: on page 15, on how to choose an object to program into the software using the camera, the image used as an example is a photo of a young child with his face blurred, wearing a headband with the effigy of ISIS, in a shooting stance and holding a real firearm.

By relating learning content to its usefulness in military training or jihad [1], the *ISIS-Scratch* textbook most certainly contributes to students' military and religious indoctrination. Momanu [37] explains that the educational system, from which the curriculum and educational materials such as textbooks are derived, is a tool of indoctrination when the learning content is inseparable from the state ideology. It appears clear that the *ISIS-Scratch* textbook is indeed an instrument of indoctrination in which the learning contents related to programming are subordinated to contextual passages of a military and religious nature (or, as in the "Dinar Challenge," more political in nature, which in the case of ISIS is inseparable from its religious nature), and where an iconography strongly marked by militarism is associated with the learning activities, even when the activity itself is not military or religious in nature. Militarism and religion are also intertwined, as three-quarters of the activities with a military context are introduced with a surah from the Qur'an.

As for the iconography in the textbook, a significant proportion of images are military or religious in nature (Fig 8). Of the 393 *foreground* images (Fig 9), which are representations of Scratch programming objects and images used to decorate the page, 50.64% are military in

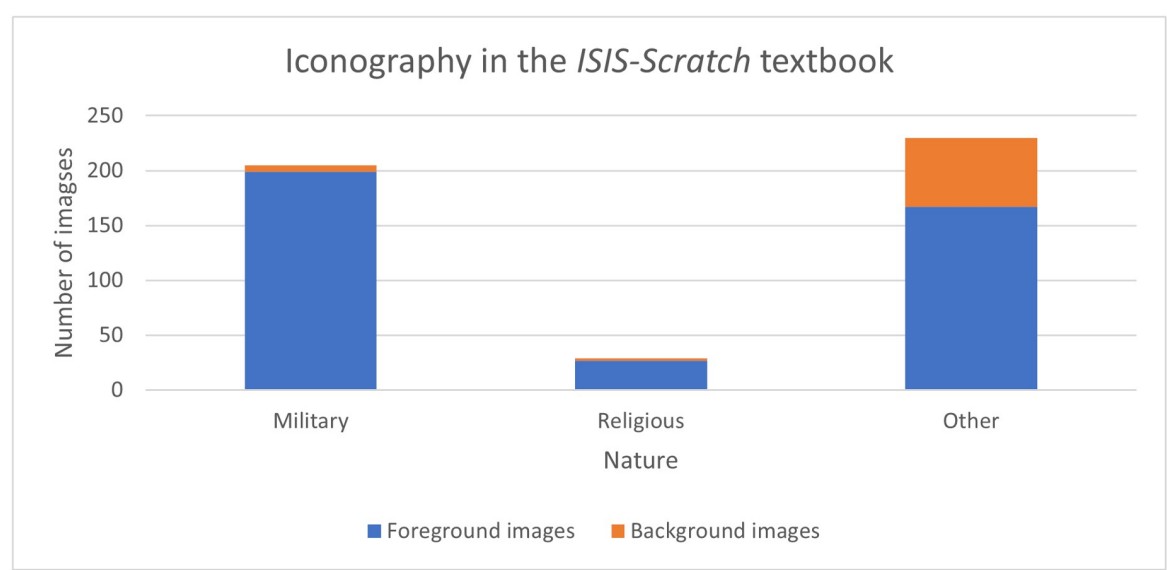

**Fig 8. Iconography in the *ISIS-Scratch* textbook.**

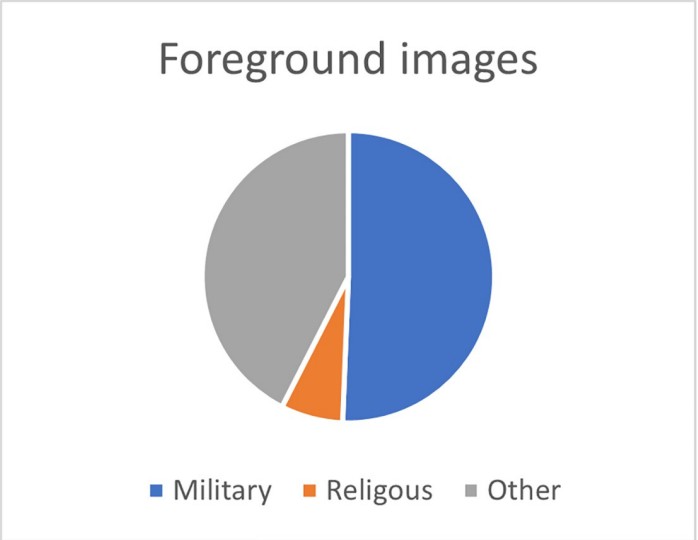

**Fig 9. Repartition of foreground images according to their nature.**

nature (including images of soldiers from ISIS, Iraq, or the US, various weapons, and military vehicles) and 6.87% are religious in nature (mostly pictures of a child praying in the "Learning to Pray" exercise). Of the 71 *background* images (Fig 10), just over 11% are military or religious in nature, and a majority (76.06%) are scenery (tree, city, desert, ocean background, rock, etc.), which serve as background for the foregrounded Scratch objects.

As an example, on page 39 (Fig 11), foreground images that are of "military" nature (ISIS soldiers and a tank) are superimposed on a background image (a forest) coded as being of "other" nature. The details of the iconographic analysis can be found in Table 3: Iconography in the *ISIS-Scratch* textbook.

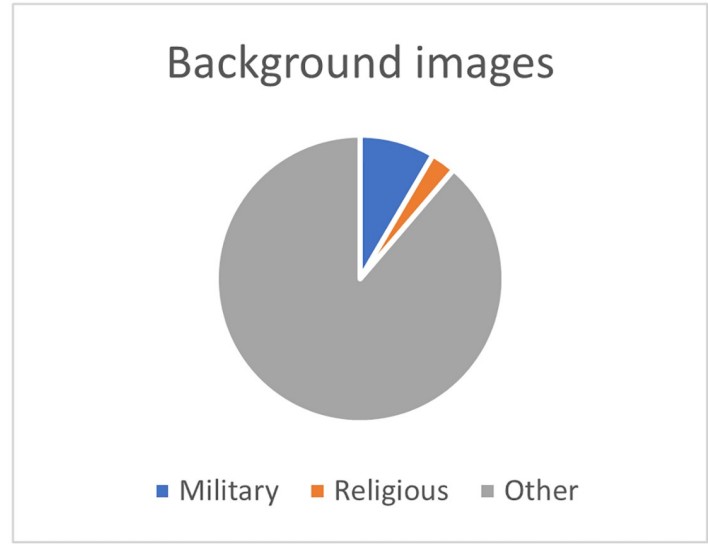

**Fig 10. Repartition of background images according to their nature.**

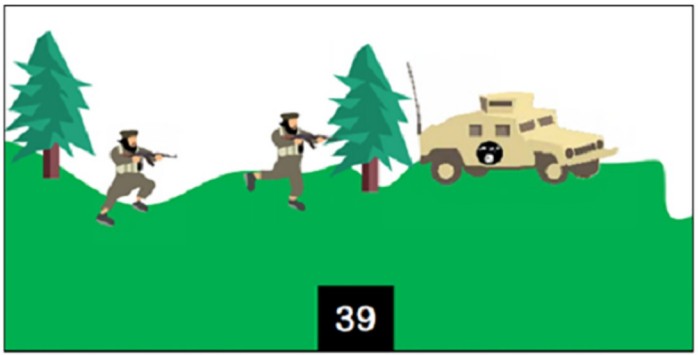

**Fig 11. ISIS soldiers and tank (foreground) in a forest (background) [18].**

**Table 3. Iconography in the *ISIS-Scratch* textbook.**

| Nature | Image | Number of foreground images | % of grand total | Number of background images | % of grand total |
|---|---|---|---|---|---|
| **Military** | Firearm | 29 | 7.38 | 4 | 5.63 |
| | Knife (saber) | 1 | 0.25 | | 0.00 |
| | Military vehicle (plane, truck, tank, helicopter, submarine) | 32 | 8.14 | 2 | 2.82 |
| | Military base | 18 | 4.58 | 0 | 0.00 |
| | Bomb | 3 | 0.76 | 0 | 0.00 |
| | Child with gun | 1 | 0.25 | 0 | 0.00 |
| | Explosion | 6 | 1.53 | 0 | 0.00 |
| | Man (dead) | 2 | 0.51 | 0 | 0.00 |
| | ISIS soldier | 44 | 11.20 | 0 | 0.00 |
| | National army soldier (Iraq or Syria) (dead) | 20 | 5.09 | 0 | 0.00 |
| | National army soldier (Iraq or Syria) (alive) | 38 | 9.67 | 0 | 0.00 |
| | US soldier | 5 | 1.27 | 0 | 0.00 |
| | **Total** | **199** | **50.64** | **6** | **8.45** |
| **Religious** | Adult man praying | 6 | 1.53 | 0 | 0.00 |
| | ISIS flag | 1 | 0.25 | 0 | 0.00 |
| | Iraq flag | 6 | 1.53 | 0 | 0.00 |
| | Child praying | 11 | 2.80 | 0 | 0.00 |
| | Mosque | 3 | 0.76 | 2 | 2.82 |
| | **Total** | **27** | **6.87** | **2** | **2.82** |
| **Other** | Animal | 33 | 8.40 | 4 | 5.63 |
| | Scenery (tree, desert, forest, building, brick wall, city, etc.) | 41 | 10.43 | 54 | 76.06 |
| | Geometric shape | 20 | 5.09 | 4 | 5.63 |
| | Man (alive) | 8 | 2.04 | 0 | 0.00 |
| | Computer | 9 | 2.29 | 0 | 0.00 |
| | Everyday object | 56 | 14.25 | 1 | 1.41 |
| | **Total** | **167** | **42.49** | **63** | **88.73** |
| **Grand total** | | **393** | **100.00** | **71** | **100.00** |

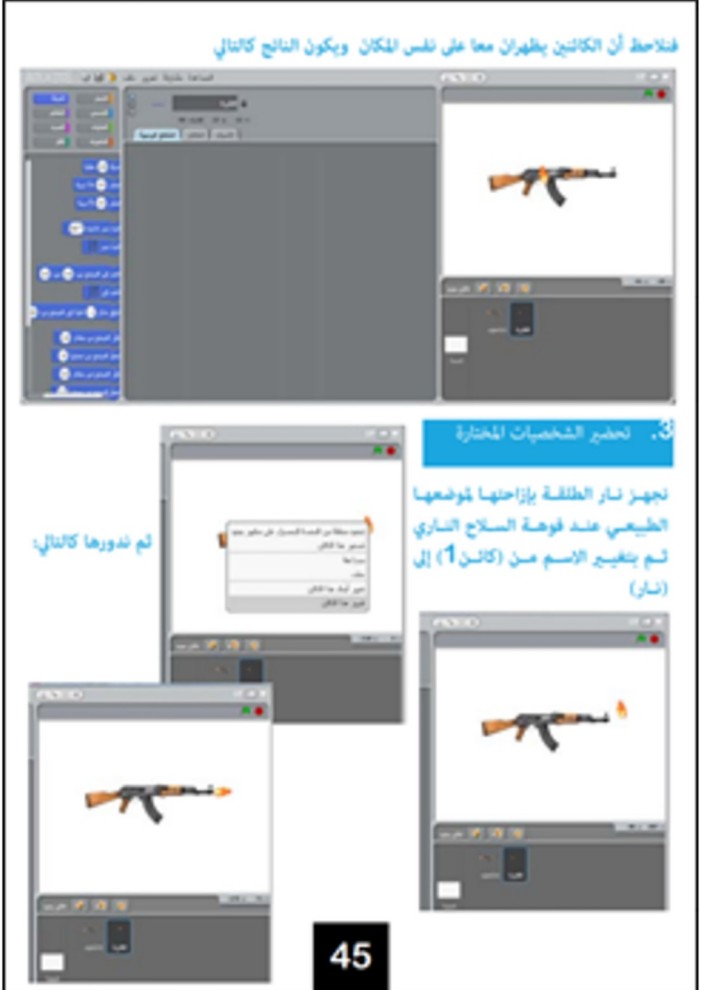

**Fig 12. Kalashnikov Challenge [18].**

The elements of military and religious indoctrination in the iconography and contexts used for the *ISIS-Scratch* textbook's learning activities are evident. These elements take on a concrete meaning when, in several activities, students have to program actions or situations associated with war, such as reproducing the sound of a gunshot and the flames coming out of the barrel (Fig 12) or shooting an enemy soldier and representing his death in a very explicit way (including dismemberment, blood, etc.). Military images are sometimes added to activities that are not put in a military context. For example, in the activity "Moving the car" [18], which has no contextual text (a statement of the learning to be done serves as an introduction), the vehicle that must be programmed to move is a tank displaying the ISIS flag. Military-oriented additions are also part of the "Farm Challenge" [18], where apples falling from a tree must be caught in a basket, taking care not to catch hidden bombs.

The indoctrination elements in the textbook contribute to instilling new values and attitudes in students [37], including militarization of the learning subject and hatred of the enemy [39]. The enemy is then a foe against whom violence is justified as an act of self-defense. Throughout the textbook, Western states and Iraq/Syrian armies are represented as supreme enemies. They are very often armed and dangerous-looking, and are often shot down (one-

third of the Iraqi/Syrian soldiers represented in the textbook are dead), which contributes to the militarization of the learners. In this way, ISIS promotes a worldview consistent with its military and religious objectives. In this sense, the final activity in the textbook, the "Dinar Challenge," is a good example of how ISIS incorporates elements of indoctrination into the textbook while fostering a worldview in students that gives young people a moral basis for justifying the use of extreme violence [1].

## 5. Conclusion

ISIS quickly established a relatively robust and well institutionalized educational system, within which it is easy to recognize the dominant ideology of the totalitarian regime, namely an action-oriented (toward jihad) Salafi doctrine. Based on our *Introduction to Programming with Scratch* textbook analysis and based on the theoretical frameworks surrounding curricula, textbooks, religious/military indoctrination, and programming learning, we can formulate observations on ISIS's programming curriculum, specifically on its pedagogical intentions, its actual curriculum, and its indoctrination value.

First, it is clear that the intentions of the ISIS programming curriculum are more about religious and military injunctions to build the caliphate than they are about developing 21st-century skills such as computational thinking. Despite some passages in the textbook referring to international trends in teaching and learning programming, it is evident that the ISIS programming curriculum is instrumentalized within an educational and social agenda marked by the religious and military ideology promoted by ISIS.

Second, although the progression of learning in the sequence of activities designed by ISIS seems logical and, overall, well-ordered, the ISIS programming curriculum places too little emphasis on the higher levels of Bloom's taxonomy as applied to programming and focuses more on the memorization of concepts and the reproduction of predetermined codes than on the acquisition of computational thinking. Thus, the curriculum does not encourage the development of 21st-century skills such as problem solving, discovery learning, or creativity—nor for that matter, the transfer of programming knowledge to different contexts. ISIS seems to have considered the potential of having young people learn programming (especially the "usefulness" aspect of programming pertaining to ISIS's online recruitment and propaganda capacity) but seems to undervalue the pedagogical and didactic aspects specific to this discipline.

Finally, the analysis of the iconography and contextual elements used for the learning activities shows that the *ISIS-Scratch* textbook is particularly rich in elements of military and religious indoctrination. More than a third of the activities involve military or religious contextual elements, and over half of the foreground images are military in nature. Given that illustrations create more persistent images and are more likely to foster prejudice than written text in the minds of students by appealing to their emotions [32], we consider that the textbook's iconography, superimposed on the military or religious narrative of many exercises, effectively participates in the indoctrination of students by helping to inculcate values consistent with ISIS's jihadist ideology.

This contribution seeks to better understand ISIS's approach to education, including its methods, intentions, and goals. It is important to note, however, that the results obtained are difficult to generalize because the corpus analyzed and the context surrounding the ISIS educational reform are extreme cases. Interpretations from these results are only applicable to the learning and teaching of programming under the ISIS educational reform from 2014 to 2017. Nevertheless, these results remain relevant to reconstruction initiatives that aim at rebuilding the impacted educational systems, as well as to curriculum development in technology,

specifically programming. Furthermore, we hope that this study can contribute to the de-indoctrination and demilitarization efforts deployed for the impacted youth and groups.

## Author Contributions

**Conceptualization:** Marion Deslandes-Martineau, Patrick Charland.

**Formal analysis:** Marion Deslandes-Martineau, Patrick Charland, Hugo G. Lapierre.

**Funding acquisition:** Chirine Chamsine.

**Investigation:** Marion Deslandes-Martineau, Patrick Charland.

**Methodology:** Marion Deslandes-Martineau, Patrick Charland, Hugo G. Lapierre.

**Validation:** Marion Deslandes-Martineau, Patrick Charland, Olivier Arvisais, Vivek Venkatesh, Mathieu Guidère.

**Visualization:** Chirine Chamsine.

**Writing – original draft:** Marion Deslandes-Martineau, Patrick Charland, Hugo G. Lapierre.

**Writing – review & editing:** Marion Deslandes-Martineau, Patrick Charland, Hugo G. Lapierre, Olivier Arvisais, Chirine Chamsine, Vivek Venkatesh, Mathieu Guidère.

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
