## [Decision Letter · Decision Letter 0]

26 Jul 2021

PONE-D-21-14928

The programming curriculum within ISIS

PLOS ONE

Dear Authors,

Thank you for submitting your manuscript to PLOS ONE. After careful consideration, we feel that it has merit but does not fully meet PLOS ONE’s publication criteria as it currently stands. Therefore, we invite you to submit a revised version of the manuscript that addresses the points raised during the review process.

We look forward to receiving your revised manuscript.

Kind regards,

Marcel Pikhart

Academic Editor

PLOS ONE

2.  We note that Figures 1-11 in your submission contain copyrighted images. All PLOS content is published under the Creative Commons Attribution License (CC BY 4.0), which means that the manuscript, images, and Supporting Information files will be freely available online, and any third party is permitted to access, download, copy, distribute, and use these materials in any way, even commercially, with proper attribution. For more information, see our copyright guidelines: http://journals.plos.org/plosone/s/licenses-and-copyright.

a. You may seek permission from the original copyright holder of Figures 1-11 to publish the content specifically under the CC BY 4.0 license.

Additional Editor Comments (if provided):

Reviewers' comments:

Reviewer's Responses to Questions

**Comments to the Author**

1. Is the manuscript technically sound, and do the data support the conclusions?

Reviewer #1: Yes

Reviewer #2: Yes

2. Has the statistical analysis been performed appropriately and rigorously? 

Reviewer #1: Yes

Reviewer #2: N/A

3. Have the authors made all data underlying the findings in their manuscript fully available?

Reviewer #1: Yes

Reviewer #2: Yes

4. Is the manuscript presented in an intelligible fashion and written in standard English?

Reviewer #1: Yes

Reviewer #2: Yes

5. Review Comments to the Author

Reviewer #1: The paper describes the education curriculum in ISIS, with a focus on programming and the Scratch programming language. The paper is well written. I commend the authors for using examples from the textbook, incorporating both iconography and contexts, demonstrating how ISIS's reinforces specifically in their programming curriculum its indoctrination values. The comparison of youth programming curricula through Bloom's taxonomy further emphasized the lack of developing higher-order skills in their education system making easier prey for indoctrination.

Some minor comments:

I believe that the keyword are a little off, especially "computer science" and "programming", the authors should, by the very least, add the word "education" to each of these keywords.

"...the ordering of these appears consistent with a classical ordering, such as that proposed by " ext-link-type="uri" xlink:type="simple">code.org." Can the authors elaborate more about code.org?

"This textbook being the only ISIS textbooks" - textbook?

The paper could encourage a study on the curriculum in other Islamic authorities.

The authors correctly refer to ISIS as "a terrorist political organization of Salafist jihadist ideology" in the introduction, but I believe that this reference should also be made in the abstract.

Reviewer #2: 1) There is an error in the enumeration of sections in the article, as the same enumeration is repeated in two different sections, see line 320 and 352. (2.4 Military indoctrination and 2.4 Overview of the scientific literature on ISIS textbooks). It is advisable to change the enumeration.

2) A data analysis section should be included, describing how the data analysis was conducted.

3) It should be explained whether the content analysis is manual or software-based. In the latest case, explain the software used.

4) It should be included what type of content analysis processes have been carried out, for example: in the results section there are tables with percentages, but it is not previously explained that a percentage analysis of activities or images is carried out.

5) It is advisable to draw up a category tree specifying the categories and codes used in the content analysis.

6) The bibliography should be updated, as only 37% is from after 2015.

6. PLOS authors have the option to publish the peer review history of their article (what does this mean?). If published, this will include your full peer review and any attached files.

Reviewer #1: No

Reviewer #2: No

---

## [Author Response · Author response to Decision Letter 0]

24 Sep 2021

The revisions applied to the manuscript after the first decision letter are detailed in the document 'Response to reviewers'.

See the comments section for the response to the second revision.

---

## [Decision Letter · Decision Letter 1]

8 Mar 2022

The Programming Curriculum within ISIS

PONE-D-21-14928R1

Dear Authors,

We’re pleased to inform you that your manuscript has been judged scientifically suitable for publication and will be formally accepted for publication once it meets all outstanding technical requirements.

Kind regards,

Marcel Pikhart

Academic Editor

PLOS ONE

Additional Editor Comments (optional):

Reviewers' comments:

Reviewer's Responses to Questions

**Comments to the Author**

1. If the authors have adequately addressed your comments raised in a previous round of review and you feel that this manuscript is now acceptable for publication, you may indicate that here to bypass the “Comments to the Author” section, enter your conflict of interest statement in the “Confidential to Editor” section, and submit your "Accept" recommendation.

Reviewer #1: All comments have been addressed

Reviewer #2: All comments have been addressed

2. Is the manuscript technically sound, and do the data support the conclusions?

Reviewer #1: Yes

Reviewer #2: Yes

3. Has the statistical analysis been performed appropriately and rigorously? 

Reviewer #1: N/A

Reviewer #2: Yes

4. Have the authors made all data underlying the findings in their manuscript fully available?

Reviewer #1: No

Reviewer #2: Yes

5. Is the manuscript presented in an intelligible fashion and written in standard English?

Reviewer #1: Yes

Reviewer #2: Yes

6. Review Comments to the Author

Reviewer #1: The authors have addressed my concerns and I am happy to advocate acceptance. There is no need for any further revision.

Reviewer #2: The authors of this research have corrected all the aspects pointed out in the evaluation process of the article.

7. PLOS authors have the option to publish the peer review history of their article (what does this mean?). If published, this will include your full peer review and any attached files.

Reviewer #1: No

Reviewer #2: No

---

## [Editor Report · Acceptance letter]

6 Apr 2022

PONE-D-21-14928R1 

The Programming Curriculum within ISIS 

Dear Dr. Deslandes-Martineau:

I'm pleased to inform you that your manuscript has been deemed suitable for publication in PLOS ONE. Congratulations! Your manuscript is now with our production department. 

Kind regards, 

on behalf of

Dr. Marcel Pikhart 

Academic Editor

PLOS ONE